# Effect of Dietary Supplementation of *Bacillus subtilis* on Growth Performance, Organ Weight, Digestive Enzyme Activities, and Serum Biochemical Indices in Broiler

**DOI:** 10.3390/ani12121558

**Published:** 2022-06-16

**Authors:** Taha M. Mohamed, Weizhong Sun, Gifty Z. Bumbie, Waleed M. Dosoky, Zebin Rao, Ping Hu, Liuting Wu, Zhiru Tang

**Affiliations:** 1Laboratory for Bio-Feed and Molecular Nutrition, College of Animal Science and Technology, Southwest University, Chongqing 400715, China; Tahaabdelameed@alexu.edu.eg (T.M.M.); swz2012@swu.edu.cn (W.S.); giftyziema@gmail.com (G.Z.B.); raozebin2020@163.com (Z.R.); huping0913@163.com (P.H.); 17752785935@163.com (L.W.); 2Department of Animal and Fish Production, Faculty of Agriculture (Saba Basha), Alexandria University, Alexandria 21531, Egypt; W-dosoky@alex.edu.eg

**Keywords:** *Bacillus subtilis*, growth performance, organ weight, blood indices, digestive enzyme activity

## Abstract

**Simple Summary:**

Broiler chickens have been consumed widely in many countries around the world because they can fulfill the nutritional needs of humans from meat. Antibiotics have been used in the broiler diet to reduce poultry pathogens and promote growth performance, but the overuse of antibiotics in the poultry industry has led to serious consequences for public health. Hence, probiotics are a safe and healthy alternative to antibiotics. *Bacillus subtilis* is a common probiotic bacteria supplement formulated as a healthy probiotic strain. This study aimed to investigate the effects of dietary supplementation of *Bacillus subtilis* on growth performance, relative organ weights, and serum biochemical indices.

**Abstract:**

This study was conducted to investigate the effects of supplementing *Bacillus subtilis* and an antibiotic (Zinc bacitracin) in the diet of broilers on growth performance, organ weight, blood metabolites, and digestive enzymes of broiler chickens. A total of 600 1-d Arbor Acres broilers were randomly allotted to five treatments. Each treatment consisted of six replicates with four pens, and each pen had five birds. The chicks were fed (1) the basal diet (control), (2) the basal diet with 500 mg/kg Zinc bacitracin (APZ), (3) the basal diet with *B. subtilis* at 1 × 10^8^ CFU/g (*B.Sut*-1), (4) the basal diet with *B. subtilis* at 3 × 10^8^ CFU/g (*B.Sut*-3), and (5) the basal diet with *B. subtilis* at 5 × 10^8^ CFU/g (*B.Sut*-5). The experiment lasted for 42 days. In this study, the supplementation of diets with *B. subtilis* (*B.Sut*-3 and *B.Sut*-5 groups) increased body weight gain from 1 to 21 days compared with control (*p* < 0.05). Additionally, the *B.Sut*-3 group had a significantly heavier bursa of Fabricius than control at 21 days (*p* < 0.05). Serum total protein, albumin, and high-density lipoprotein concentrations were increased in *B.Sut*-5 and APZ groups (*p* < 0.05) over the whole period. Serum low-density lipoprotein, very low-density lipoprotein, triglyceride, and total cholesterol concentrations were decreased in *B.Sut*-5 and APZ groups at 21 and 42 days (*p* < 0.05). Chicks in the *B.Sut*-5 and APZ groups had higher serum lipase, pepsin, and amylase activities (*p* < 0.05) at 21 and 42 days. From the results obtained from the study, it can be concluded that *Bacillus subtilis* ATCC19659 at 5 × 10^8^ CFU/g could be applied as an alternative to antibiotics in poultry diets.

## 1. Introduction

Poultry production is considered one of the important fields in animal production in the world. Poultry helps bridge the food gap because poultry provides humans with meat and eggs, which are important sources of animal protein [1]. In addition, poultry meat has high nutritional value, in addition to being relatively cheaper. Moreover, poultry meat has a small amount of fat with a high portion of unsaturated fatty acids and a low cholesterol level. Additionally, it contains a percentage of niacin, which is proven to scientifically increase the levels of high-density lipoprotein (HDL) cholesterol. Broiler chickens are raised for a period of 6 to 7 weeks. Broiler chickens are reared in almost all parts of the world and processed for breast meat, wings, and thighs. In recent times, poultry products have increased in demand around the world [2]. Climate change negatively affects the production of field crops, such as corn, soybeans, and the rest of the components that go into the ingredients of poultry diets. Feed is one of the biggest items of cost in poultry production and it alone accounts for 65% of the whole of poultry production. The increase in expenditure on feed ingredients is resulting in fewer earnings for poultry farmers. Feed additives and nutritional supplements are of great importance in the poultry industry and healthcare systems due to their beneficial impacts, such as stimulating growth and immunity enhancement. 

One antibiotic has been used for a long time in poultry production—since 1951. Zinc bacitracin is used as a growth promoter as well as in some topical preparations in veterinary medicine. The Food and Drug Administration (FDA) in the USA approved the use of antibiotics in chicken and turkey production for a long time Recently, many European Union countries and the USA have banned the use of antibiotics in livestock and poultry feed [3] due to the dissemination of antibiotic-resistant strains of pathogenic and non-pathogenic organisms into the environment and their further transmission to humans via the food chain [4,5]. Therefore, some countries are working to develop a safe and healthy alternative to antibiotics in livestock and poultry feed [6]. Many of the previous research works have highlighted the importance of probiotic microorganisms as an alternative to antibiotics. Recently, the use of probiotics in poultry diets has increased significantly due to the increase in demand for antibiotic-free poultry.

*Bacillus subtilis* is a forming spore that survives in the environment during harsh conditions, which can be resistant to alkali, acid, and heat. Therefore, *Bacillus subtilis* has the characteristics that make it an alternative to antibiotics, such as the ability of spores to grow fast in the intestinal tract. *Bacillus subtilis* grows in an aerobic condition, so it takes a high amount of free oxygen in the intestinal tract; it can therefore restrict the growth of pathogenic aerobic bacteria and enhance the growth of anaerobic bacteria, such as yeast, *Lactobacillus,* and *Bifidobacterium* [7,8], which leads to the promotion and development of the intestinal function. Dietary supplementation with *Bacillus subtilis* improves broiler performance and carcass quality in broiler chickens [9,10,11]. Furthermore, *Bacillus subtilis* can produce exogenous digestive enzymes [12,13], which leads to an improved benefit from the digestion of nutrients. Nevertheless, the best concentration for probiotic inclusion may depend on the bacteria strain, so an increased inclusion rate does not always result in improved results [14,15]. Yet, to our knowledge, information is lacking about the effects of *Bacillus subtilis* ATCC19659 on growth, organ weight, blood biochemical indices, and digestive enzyme activities of broiler chickens. Furthermore, we hypothesized that *Bacillus subtilis* ATCC19659 has beneficial effects on broiler chickens. Therefore, this study was designed to assess the effectiveness of *Bacillus subtilis* ATCC19659 on the growth performance, organ weight, blood biochemical indices, and digestive enzyme activities of broiler chickens. This will provide scientific foundation for the use of antibiotic substitutes.

## 2. Materials and Methods

### 2.1. Bacterial Strain

*Bacillus subtilis* ATCC19659 was provided by (KWIk-STIk™, Microbiologics, Microbiologics, Inc., Saint Cloud, MN, USA). The strain of bacteria was cultivated in a nutrient medium (g/L: Beef Extract 5; Peptone 3; pH medium 6.8 ± 2) and placed in a shaker–incubator (200 rpm) at 37 °C for 14 h in an aerobic environment following the method of Dumitru et al. [16,17]. The inoculum was measured by ten-fold serial dilutions using phosphate-buffered saline solution (PBS), and then, 1 mL from 10^−5^ to 10^−10^ was incubated on nutrient agar medium (g/L (g/L: Beef Extract 5; Peptone 3; bacteriological agar 15; distilled water). To estimate the viability of spores forming, the vegetative cell was eliminated by subjection to thermal treatment (120 °C, 10 min). Serial dilutions in PBS on agar nutrient medium were incubated at 37 °C in aerobic conditions for 14 h. The bacterial pellets were collected by centrifugation (5000 rpm, 20 min, 4 °C), washed three times, and then resuspended in PBS solution. Finally, the total content from the strain was prepared of a viable bacterium. Every morning, before the inclusion of *Bacillus subtilis* ATCC19659 into broiler diets, all birds had their water and feed withdrawn for about 3 h. After this period, the feed was mixed with various concentrations of *Bacillus subtilis* ATCC19659, and the birds consumed it within 30 min. Furthermore, the birds in the control and APZ groups were fed the same quantity of their customary food. After the feed was completed, the birds were given their regular water and feed. The final concentrations of *Bacillus subtilis* ATCC19659 in the feed were 1, 3, and 5 × 10^8^ CFU/g following the method of [18].

### 2.2. Birds, Housing, Diets, and Experimental Design

A total of 600 1-d Arbor Acres broilers were randomly allotted to 5 treatments. Each treatment consisted of 6 replicates with 4 pens, and each pen had 5 birds. The chicks were fed (1) the basal diet (control), (2) the basal diet with 500 mg/kg Zinc bacitracin (APZ), (3) the basal diet with *B. subtilis* at 1 × 10^8^ CFU/g (*B.Sut*-1), (4) the basal diet with *B. subtilis* at 3 × 10^8^ CFU/g (*B.Sut*-3), and (5) the basal diet with *B. subtilis* at 5 × 10^8^ CFU/g (*B.Sut*-5). The experiment lasted for 42 days. All birds were kept in stainless-steel pens of identical size (30 cm × 175 cm × 155 cm). Room temperature was 33 °C in the first three days, and every week, it was gradually reduced by 3 °C to reach 24 °C until the end of the experiment. The basal diet was calculated to meet NRC [19] requirements during the starter (1–21 days) and finisher period (22–42 days). The feed composition and chemical analysis of the experimental diets used in this study are shown in Appendix A. All the birds were kept under artificial light in addition to being allowed ad libitum access to feed and water throughout the experimental period. At day 14, birds in all groups were vaccinated through their drinking water against essential Newcastle disease using (LaSota B1 Strain of Newcastle disease virus in live freeze-dried form YEBIO^®^). The present experiment was conducted at the experimental poultry house of the College of Animal Science and Technology, Southwest University, China. Birds were handled according to the instructions described by the Animal Care Committee of Chongqing Province, P.R. China (IACUC-20190824-24).

### 2.3. Growth Performance

The growth performance of the birds was measured at 1, 21, and 42 days of the experiment to calculate the body weight gain, feed intake (FI), and feed efficiency ratio (FER). The feed intake of each replicate was calculated as the amount of residual from feed fed to the birds on a pen-to-pen basis. The feed efficiency ratio (FER) was calculated by dividing the average body weight gain by the feed intake. At the end of the experiment, the performance index was calculated according to the following equation: performance index = (feed efficiency ratio (FER) × body weight gain).

### 2.4. Blood Collection and Carcass Traits

Blood samples were collected at 21st days (1 bird per replicate) and 42nd days (2 birds per replicate). Birds with body weights close to the mean body weight from each treatment were selected from two periods (21st and 42nd days). Blood samples were collected from the jugular vein in a 10 mL tube individually. The samples were centrifuged at 3000 rpm for 20 min at 4 °C, and the serum was stored at −20 °C for assaying. After blood sampling, birds were euthanized by an intravenous injection of sodium pentobarbital (50 mg/kg BW), exsanguinated, scalded in hot water (60 °C for 45 s) after bleeding, and then defeathered. Carcasses were eviscerated manually and weighed. Liver, gizzard, spleen, heart, pancreas, and bursa were removed and weighed separately, and the weights of these organs were expressed as a percentage of live body weight.

### 2.5. Biochemical Analysis

Total protein, albumin, triglyceride, high-density lipoprotein (HDL), low-density lipoprotein (LDL), very low-density lipoprotein (VLDL), total cholesterol (T. cholesterol), as well as creatine, uric acid, alanine aminotransferase (ALT), aspartate aminotransferase (AST), alkaline phosphates (ALP), pepsin, lipase, and amylase in the serum were measured by chicken-specific ELISA kits. Kits were obtained from Nanjing Jiancheng Bioengineering Institute, Nanjing, China. The kit procedure was carried out according to the manufacturer’s instructions. The information of all kits is shown in Appendix A.

### 2.6. Statistical Analysis

All data were statistically analyzed using one-way ANOVA. It was performed using the SAS program (SAS Institute Inc., Cary, NC, USA). Duncan’s test (*p* < 0.05) was used when significant differences were found among the treatment groups. The mean ± SEM were used to express the data.

## 3. Results

### 3.1. Growth Performance

The effects of *B. subtilis* as a dietary supplement on growth performance of broilers at different periods are shown in Table 1. From 1 to 21 days, both *B.Sut*-3 and *B.Sut*-5 groups experienced increased body weight gain compared with the control group (*p* < 0.05), while no significant difference was observed among all dietary treatment groups in body weight gain, feed intake, feed efficiency ratio in all the different periods. Equally, the performance index did not record any significant difference among all treatment groups at the end of the experiment.

### 3.2. Carcass Measurements

The effects of supplementary *B.subtilis* and APZ on the relative organ weight of broiler chickens are shown in Table 2. The relative weights of all organs (proventriculus, liver, gizzard, abdominal fat, spleen, heart, bursa, and pancreases) remained unaffected by the supplementation of B. subtilis and the APZ dietary supplement at 21st and 42nd days. except for the relative weight of the bursa of Fabricius was increased in the *B.Sut*-3 group at 21 days (*p* < 0.05) ( Table 2).

### 3.3. Biochemical Blood Indices

The effects of dietary APZ and *B. subtilis* at 21st and 42nd days on the serum biochemical variables of broilers are shown in Table 3. Serum concentrations of total protein, albumin, and HDL of birds belonging to the *B.Sut*-5 and APZ groups were significantly increased at 21st and 42nd days in comparison with the control group (*p* < 0.05), as shown in Table 3. Serum LDL, VLDL, triglyceride, and total cholesterol were reduced in the *B.Sut*-5 and APZ groups at 21st and 42nd days in comparison with the control group (*p* < 0.05) (Table 3).

### 3.4. The Kidney and Liver Functions

The effect of dietary supplementation with *B. subtilis* and APZ on the functions of the kidney and liver on the 21st and 42nd days are shown in Table 4. APZ and *B.Sut*-5 groups were significantly lower serum creatinine and uric acid than the control group at 21st and 42nd days. However, serum ALT and AST were not affected in broiler among all treatments in the same period. However, serum alkaline phosphates (ALB) increased in APZ and *B.Sut*-5 groups when compared with the control in the same two periods (*p* < 0.05) (Table 4).

### 3.5. Serum Digestive Enzyme Activities

The results of dietary administration of *B. subtilis* on the activities of the digestive enzyme are shown in Table 5. The activities of the digestive enzymes (pepsin, amylase, and lipase) in both APZ and *B.Sut*-5 groups were significantly increased as compared with the control group in the two periods (*p* < 0.05) (Table 5).

## 4. Discussion

At the present time, many probiotic products are widely used in several commercial applications. Probiotics such as *Bacillus subtilis* could gain importance as an alternative antibiotic in poultry diets because *Bacillus species* are nonpathogenic and Gram positive [9,20].

Probiotic dietary supplementation in broilers has a positive effect on growth performance. In this way, adding different levels of *B. subtilis* showed a significant increase in body weight gain throughout the trial period [9]. Zhang et al. [21] reported an increase in the average daily gain of broilers when feed was supplemented with *Bacillus* spp. (10^5^ and 10^8^ cfu/kg). Additionally, the growth performance of broiler chickens was improved by supplementation of *Bacillus species* [22,23]. Amerah et al. [24] demonstrated that the supplementation of a diet with *B. subtilis* at 1.5 × 10^8^ cfu/kg for broilers significantly affected the average daily gain among the treatment groups. In this present study, the average daily weight gain was increased in the *B.Sut*-3 and *B.Sut*-5 groups from 1 to 21 days.

Bursa of Fabricius is an important site for the maturation of T and B lymphocytes, and the size and mass of the bursa of Fabricius are very important for giving general information about the immune system in birds [25]. In this present study, the dietary inclusion of *B.*
*subtilis* ATCC19659 (*B.Sut*-3 group) increased the bursa of Fabricius weight at 21 days, unlike the other organs, which did not show significant difference among treatments. The result is consistent with Park and Kim [25] found that the relative weights of bursa of Fabricius of bird fed with B. subtilis B2A diets were significantly increased when compared with the control.. Probiotics added as dietary supplementation increased the bursa of Fabricius compared with the control [26,27,28,29]. Alkhalf et al. [30] also reported that probiotic supplemented in broiler chicks’ diet increased immune organ weights compared with the control. Reis et al. [31] documented that feeding *Bacillus subtilis* (DSM 17299) could not influence the relative weight of liver and spleen. Ciurescu et al. [11] showed that the addition of *Bacillus subtilis* ATCC 6051 to the broiler diet did not affect the weight of the heart, liver, gizzard, and pancreas. However, Zhang et al. [32] illustrated that dietary addition of *B. subtilis* did not affect the weight of bursa of Fabricius at the end of the experiment. Additionally, Hatab et al. [10] found that probiotic as dietary supplementation increased the relative weight of organs (heart, kidney, proventricular, thymus, and liver).

In our results, some blood parameters were influenced with the dietary *B. subtilis* supplementation at 21st and 42nd days. Dietary inclusion of *B. subtilis* (B.Sut-5) and antibiotic (APZ) increased serum total protein, albumin, and alkaline phosphates. *B. subtilis* improves the usage of dietary protein by suppression of pathogen growth, which decreases the breakdown of protein into nitrogen (N2), thereby reducing dietary protein efficiency and increasing the rate of nutrient absorption from the surface [12,33,34,35]. Li et al. [36] found that ALP increased with addition of *Bacillus subtilis* in broiler diets. Consistent with our results, Hatab et al. [10] who reported that here was no effect on serum albumin and total protein by the addition of probiotic. however, alkaline phosphatase, ALT, and AST were significantly decreased with dietary inclusion of probiotic. Additionally, dietary probiotic supplementation did not influence the concentration of albumin and total protein when compared with the control [37,38].

Presently, the inclusion of *B. subtilis* (*B.Sut*-5) and (APZ) in broiler diets decreased serum LDL, VLDL, triglyceride, and total cholesterol more than the control group over the whole period. The assimilation of cholesterol is one way of another possible mechanism of biological bacteria supplementation (*B. subtilis*). It can produce active bile salt hydrolase and maintain bile salt homeostasis, so it sometimes needs to synthesize more bile acids to reduce the levels of cholesterol in the body pool because cholesterol is considered the precursor of bile acids [39]. Additionally, probiotics in the intestine convert cholesterol to coprostanol, which is excreted through the faeces directly [40], it inhibits the cholesterogenesis rate-limiting enzyme, or reduces the activity of hydroxy-methyl-glutarylcoenzyme-A, which is involved in cholesterol synthesis, slows down the synthesis of this steroid from acetyl-CoA [41,42]. Reduced cholesterol content in poultry products is generally linked to lower blood cholesterol levels [43]. The use of probiotics in poultry diets may result in the production of low-cholesterol eggs and meat, so probiotics can help address the growing global demand for meat and eggs that are low in saturated fat and cholesterol. Consistent with our results, the utilization of probiotics as dietary supplementation decreased triglycerides and total cholesterol [10,44,45]. Triglycerides are reduced when energy is restricted to the ability of cells to use or keep triglycerides as a source of energy, which causes lipotoxicity, the activation of inflammatory processes, and oxidative stress [42].

Creatinine and uric acid reflect the performance of kidney function. Some of the probiotic microorganisms (*B.subtilis spp.*) used creatinine, urea, uric acid, and other toxins as nutrients for growth [45]. Creatinine and uric acid were decreased in the *B. subtilis* (*B.Sut*-5) and APZ groups. Consistent with the current study, Hatab et al. [10] found that uric acid was decreased significantly by feeding probiotics. By contrast, Strompfova et al. [46] reported that there was no effect on serum uric acid levels with the addition of probiotics. However, Tonekabon et al. [47] demonstrated that dietary probiotic supplementation had higher serum uric acid than the control. In the current study, ALT and AST were not affected by the administration of *Bacillus subtilis*. Keeping AST and ALT activities within the limits of natural numbers in treatments indicated the normal status of liver function as a consequence of biological supplementation with BS-ATCC19659. The danger is in the increase in blood ALT and AST enzymes, which act as indicators of hepatocellular damage [48].

Gastrointestinal enzyme activities such as amylase, pepsin, and lipase have an important role in nutritional digestion, which improves growth performance. Presently, *B.Sut*-5 and APZ groups experienced improved pepsin, amylase, and lipase activities. Higher activity of pepsin, amylase, and lipase enhanced the digestion of protein, starch, and lipids, and this might be a possible cause for growth improvement in the current study. *Bacillus* spp. contributes to the excretion of exogenous enzymes together with producing the host from the endogenous enzymes [8,49]. Consistent with our results, Abd El-Moneim et al. [12] demonstrated dietary supplementation of *B. subtilis* spores led to increased protease, lipase, and amylase activities when compared with the control. Wang and Gu [50] also reported that dietary supplementation of *Bacillus coagulans* NJ0517 significantly increased protease and amylase activities in Arbor Acres broilers.

## 5. Conclusions

In conclusion, dietary supplementation with *Bacillus subtilis* 5 × 10^8^ CFU/g feed (*B.Sut*-5) as probiotics significantly increased body weight gain, increased the utilization of dietary protein efficiency and HDL, improved kidney function, as well as decreasing serum triglycerides and total cholesterol, LDL, and VLDL. Finally, dietary supplementation with *Bacillus subtilis* 5 × 10^8^ CFU/g feed (*B.Sut*-5) increased the production of digestive enzymes in different periods of the experiment. It could be concluded that feed additives (*Bacillus subtilis* ATCC19659) in the poultry industry might be an encouraging alternative to antibiotic growth promoters.

## Figures and Tables

**Table 1 animals-12-01558-t001:** Effect of the addition of *Bacillus subtilis* into broiler diets on broilers’ growth performance.

Items	Treatments ^1^					SEM	*p*-Value
	Control	APZ	*B.Sut*-1	*B.Sut*-3	*B.Sut*-5		
Feed intake (g)							
0–21 d	842.76	845.79	898.67	921.37	896.82	12.61	0.185
22–42 d	1665.72	1646.27	1709.91	1720.91	1678.59	17.20	0.663
1–42 d	2508.47	2492.06	2607.79	2642.28	2575.41	25.83	0.305
Weight gain (g)							
0–21 d	593.65 ^bc^	572.78 ^c^	600.32 ^ab^	610.43 ^ab^	621.93 ^a^	4.61	0.004
22–42 d	959.1	956.21	976.04	998.35	975.65	10.55	0.755
1–42 d	1552.75	1528.98	1576.36	1608.78	1597.58	11.79	0.189
Feed efficiency ratio							
0–21 d	0.70	0.68	0.67	0.67	0.7	0.01	0.606
22–42 d	0.58	0.58	0.57	0.58	0.58	0.005	0.958
1–42 d	0.62	0.61	0.61	0.61	0.62	0.004	0.642
Performance index	962.59	938.57	953.89	979.81	992.56	8.98	0.360

^1^ Control, Chicks were fed basal diet; APZ, Chicks were fed basal diet containing 500 mg zinc bacitracin/kg; *B.Sut*-1, Chicks were fed basal diet containing 1 × 10^8^ CFU *B. subtilis*/g; *B.Sut*-3, Chicks were fed basal diet containing 3 × 10^8^ CFU *B. subtilis*/g; *B.Sut*-5, Chicks were fed basal diet containing 5 × 10^8^ CFU *B. subtilis*/g. ^a,b,c^ Values in the same row with different letter superscripts mean significant differences (*p* < 0.05).

**Table 2 animals-12-01558-t002:** Effect of the addition of *Bacillus subtilis* into broiler diets on broilers’ relative organ weight at 21st and 42nd days of age.

Treatment ^1^				Items				
	Proventriculus	Liver	Gizzard	Abdominal Fat	Bursa	Spleen	Heart	Pancreases
Day 0 to 21								
Control	0.6	2.74	3.27	1.04	0.25 ^c^	0.09	0.56	0.37
APZ	0.59	2.4	3.06	1.19	0.34 ^ab^	0.09	0.56	0.31
*B.Sut*-1	0.75	2.81	3.56	1.21	0.27 ^bc^	0.08	0.65	0.33
*B.Sut*-3	0.66	2.93	3.1	1.46	0.36 ^a^	0.08	0.64	0.38
*B.Sut*-5	0.71	2.75	3.29	1.36	0.3 ^abc^	0.1	0.63	0.38
SEM	0.039	0.112	0.13	0.058	0.14	0.004	0.091	0.012
*p*-Value	0.667	0.701	0.787	0.191	0.038	0.299	0.228	0.23
Day 22 to 42								
Control	0.41	2.42	2.18	1.68	0.2	0.14	0.45	0.19
APZ	0.42	2.39	2.04	1.62	0.21	0.1	0.46	0.2
*B.Sut*-1	0.45	2.09	2.24	1.7	0.18	0.13	0.47	0.2
*B.Sut*-3	0.42	2.29	1.81	1.92	0.24	0.11	0.48	0.21
*B.Sut*-5	0.46	2.47	1.92	1.61	0.24	0.09	0.45	0.22
SEM	0.009	0.047	0.082	0.05	0.009	0.007	0.009	0.005
*p*-Value	0.459	0.076	0.442	0.292	0.139	0.152	0.797	0.283

^1^ Control, Chicks were fed basal diet; APZ, Chicks were fed basal diet containing 500 mg zinc bacitracin/kg; *B.Sut*-1, Chicks were fed basal diet containing 1 × 10^8^ CFU *B. subtilis*/g; *B.Sut*-3, Chicks were fed basal diet containing 3 × 10^8^ CFU *B. subtilis*/g; *B.Sut*-5, Chicks were fed basal diet containing 5 × 10^8^ CFU *B. subtilis*/g. ^a,b,c^ Values in the same row with different letter superscripts mean significant differences (*p* < 0.05). Data are presented as the mean ± SEM.

**Table 3 animals-12-01558-t003:** Effect of the addition of *Bacillus subtilis* into broiler diets on serum biochemical blood of broiler chickens at 21st and 42nd days old.

Treatment ^1^	Items ^2^						
	Total Protein (g/dL)	Albumin (g/dL)	Triglyceride (mg/dL)	HDL (mg/L)	LDL (mg/L)	VLDL (mg/L)	T. Cholesterol (mg/L)
Day 0 to 21							
Control	4.93 ^c^	2.57 ^d^	34.84 ^a^	10.35 ^d^	9.98 ^a^	14.19 ^a^	34.52 ^a^
APZ	6.82 ^a^	4.35 ^a^	20.1 ^d^	17.36 ^a^	5.67 ^cd^	6.85 ^c^	29.88 ^d^
*B.Sut*-1	5.38 ^bc^	3.1 ^c^	29.31 ^b^	14.01 ^c^	8.71 ^b^	10.91 ^b^	33.63 ^ab^
*B.Sut*-3	5.6 ^b^	3.3 ^c^	24.28 ^c^	13.89 ^c^	7.19 ^c^	10.23 ^b^	31.3 ^bc^
*B.Sut*-5	6.51 ^a^	3.93 ^b^	21.21 ^cd^	15.73 ^b^	6.91 ^d^	7.93 ^c^	30.58 ^d^
SEM	0.149	0.121	1.158	0.486	0.329	0.533	0.48
*p*-Value	<0.0001	<0.0001	<0.0001	<0.0001	<0.0001	<0.0001	0.002
Day 22 to 42							
Control	4.97 ^c^	2.42 ^c^	33.85 ^a^	11.03 ^c^	9.99 ^a^	14.01 ^a^	35.03 ^a^
APZ	6.69 ^a^	3.74 ^a^	23.10 ^b^	16.42 ^a^	6.08 ^c^	7.91 ^d^	30.4 ^c^
*B.Sut*-1	5.16 ^c^	2.57 ^c^	33.72 ^a^	12.36 ^c^	9.88 ^a^	11.82 ^b^	34.05 ^ab^
*B.Sut*-3	5.70 ^b^	2.98 ^b^	30.87 ^a^	14.1 ^b^	8.49 ^b^	9.7 ^c^	32.28 ^bc^
*B.Sut*-5	5.98 ^b^	3.48 ^a^	23.81 ^b^	14.86 ^ab^	7.67 ^b^	9.62 ^c^	32.15 ^bc^
SEM	0.127	0.11	0.989	0.42	0.302	0.448	0.447
*p*-Value	<0.0001	<0.0001	<0.0001	<0.0001	<0.0001	<0.0001	0.004

^1^ Control, Chicks were fed basal diet; APZ, Chicks were fed basal diet containing 500 mg zinc bacitracin /kg; *B.Sut*-1, Chicks were fed basal diet containing 1 × 10^8^ CFU/g of feed *B. subtilis*; *B.Sut*-3, Chicks were fed basal diet containing 3 × 10^8^ CFU/g of feed *B. subtilis*; *B.Sut*-5, Chicks were fed basal diet containing 5 × 10^8^ CFU/g of feed *B. subtilis*.^2^ HDL, High-density lipoprotein; LDL, Low-density lipoprotein; VLDL, Very low-density lipoprotein; T. cholesterol, Total cholesterol. ^a,b,c,d^ Values in the same row with different letter superscripts mean significant differences (*p* < 0.05). Data are presented as the mean ± SEM (*n* = 6).

**Table 4 animals-12-01558-t004:** Effect of the addition of *Bacillus subtilis* into broiler diets on serum kidney and liver functions and alkaline phosphates of broiler chickens at 21st and 42nd days old.

Treatment ^1^	Items ^2^				
	Creatine	Uric Acid	ALT	AST	ALP
	(mg/dL)	(mg/dL)	(U/L)	(U/L)	(U/100 mL)
Day 0 to 21					
Control	3.04 ^a^	2.94 ^a^	54.15	72.13	0.9 ^c^
APZ	1.48 ^c^	1.87 ^d^	55.24	74.04	1.4 ^a^
*B.Sut*-1	2.25 ^b^	2.57 ^b^	55.65	78.64	1.11 ^b^
*B.Sut*-3	2.12 ^b^	2.31 ^bc^	49.87	70.61	1.26 ^a^
*B.Sut*-5	2.03 ^b^	2.16 ^c^	62.48	76.29	1.3 ^a^
SEM	0.111	0.078	2.28	2.18	0.038
P-Value	<0.0001	<0.0001	0.558	0.81	<0.0001
Day 22 to 42					
Control	3.01 ^a^	2.9 ^a^	45.23	70.17	0.85 ^c^
APZ	2.11 ^b^	2.17 ^c^	57.4	82.24	1.25 ^a^
*B.Sut*-1	2.93 ^a^	2.91 ^a^	61.15	70.3	1.05 ^b^
*B.Sut*-3	2.44 ^b^	2.61 ^b^	52.5	79.45	1.16 ^a^
*B.Sut*-5	2.4 ^b^	2.59 ^b^	46.83	68.76	1.25 ^a^
SEM	0.083	0.063	2.237	2.294	0.032
*P*-Value	<0.0001	<0.0001	0.108	0.107	<0.0001

^1^ Control, Chicks were fed basal diet; APZ, Chicks were fed basal diet containing 500 mg zinc bacitracin/kg; *B.Sut*-1, Chicks were fed basal diet containing 1 × 10^8^ CFU/g of feed *B. subtilis*; *B.Sut*-3, Chicks were fed basal diet containing 3 × 10^8^ CFU/g of feed *B. subtilis*; *B.Sut*-5, Chicks were fed basal diet containing 5 × 10^8^ CFU/g of feed *B. subtilis*. ^2^ ALT, Alanine aminotransferase; AST, Aspartate aminotransferase; ALP, Alkaline phosphates. ^a,b,c^ Values in the same row with different letter superscripts mean significant differences (*p* < 0.05). Data are presented as the mean ± SEM (*n* = 6).

**Table 5 animals-12-01558-t005:** Effect of the addition of *Bacillus subtilis* into broiler diets on digestive enzyme activities of broiler chickens at 21st and 42nd days old.

Treatment ^1^	Items		
	Pepsin (U/mL)	Lipase (U/L)	Amylase (U/dL)
Day 0 to 21			
Control	49.04 ^b^	604.22 ^c^	154.06 ^c^
APZ	68.7 ^a^	899.63 ^a^	238.63 ^ab^
*B.Sut*-1	48.48 ^b^	671.81 ^bc^	161.77 ^c^
*B.Sut*-3	54.03 ^b^	701.04 ^bc^	190.24 ^bc^
*B.Sut*-5	63.58 ^a^	749.48 ^b^	273.08 ^a^
SEM	1.789	23.998	11.18
*p*-Value	<0.0001	<0.0001	<0.0001
Day 22 to 42			
Control	23.05 ^b^	566.4 ^bc^	137.96 ^c^
APZ	55.03 ^a^	927.49 ^a^	265.45 ^a^
*B.Sut*-1	32.81 ^b^	495.95 ^c^	113.63 ^c^
*B.Sut*-3	30.96 ^b^	654.32 ^b^	153.75 ^c^
*B.Sut*-5	35.08 ^b^	666.01 ^b^	203.69 ^b^
SEM	2.41	35.38	12.89
*p*-Value	<0.0001	<0.0001	<0.0001

^1^ Control, Chicks were fed basal diet; APZ, Chicks were fed basal diet containing 500 mg zinc bacitracin/kg; *B.Sut*-1, Chicks were fed basal diet containing 1 × 10^8^ CFU/g of feed *B. subtilis*; *B.Sut*-3, Chicks were fed basal diet containing 3 × 10^8^ CFU/g of feed *B. subtilis*; *B.Sut*-5, Chicks were fed basal diet containing 5 × 10^8^ CFU/g of feed *B. subtilis*. ^a,b,c^ Values in the same row with different letter superscripts mean significant differences (*p* < 0.05). Data are presented as the mean ± SEM (*n* = 6).

## Data Availability

The data that support the findings of this study are available on request from the corresponding author.

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
