# Peer review of "Effect of Dietary Supplementation of Bacillus subtilis on Growth Performance, Organ Weight, Digestive Enzyme Activities, and Serum Biochemical Indices in Broiler"

_animals, 2022, doi:10.3390/ani12121558_

Round 1

Reviewer 1 Report

In this study, the authors investigated the effect of supplementing broiler diets with Bacillus subtilis ATCC19659 and an antibiotic (bacitracin zinc) on growth performance, carcass characteristics, blood metabolites and digestive enzymes in broilers. There are a few more points to clarify:

1. L149, 151, and reagents from the same company can be represented together.

The fonts in Tables 1-4 are inconsistent, as are the font sizes of table comments.

3. "P < 0.05", P needs to be italicized (all these expressions are in the manuscript)

4. The units of Bacillus subtilis in the text are written in many different forms, such as CFU g-1, cfu/kg, CFU/g. Please standardize your writing.

5. The manuscript cites a large number of references from 10 years ago, and the author should cite the literature from the past 3-5 years.

6. The language of the manuscript needs professional revision.

在这项研究中,作者研究了在肉鸡饮食中补充枯草芽孢杆菌ATCC19659和抗生素(杆菌肽锌)对肉鸡生长性能,胴体特性,血液代谢物和消化酶的影响。还有一些要点需要阐明:

1. L149,151,来自同一家公司的试剂可以一起代表。

表1-4中的字体不一致,表格注释的字体大小也不一致。

3.“P<0.05”,P需要斜体(所有这些表达都在手稿中)

4.文中枯草芽孢杆菌的单位以许多不同的形式书写,例如CFU g-1,cfu / kg,CFU / g。请规范您的写作。

5、稿件引用了大量10年前的参考文献,作者应引用过去3-5年的文献。

6、稿件语言需要专业修改。

Author Response

Section: "Responses to the Comments by reviewer 1"

In this study, the authors investigated the effect of supplementing broiler diets with Bacillus subtilis ATCC19659 and an antibiotic (bacitracin zinc) on growth performance, carcass characteristics, blood metabolites and digestive enzymes in broilers. There are a few more points to clarify:

We are grateful for your acceptance of our manuscript and also for your insightful review and we hope that we have covered all the review points.

Comment 1:  L149, 151, and reagents from the same company can be represented together.

Authors'responses and locations of the revisions: Thanks. Owing to your suggestion, we modified and collected reagents together in line 167 page 4.

Comment 2:  The fonts in Tables 1-4 are inconsistent, as are the font sizes of table comments.

Authors'responses and locations of the revisions: We thank the reviewer for pointing this out. We have changed the fonts and unified the fonts in all tables.

Comment 3:  "P < 0.05", P needs to be italicized (all these expressions are in the manuscript)

Authors'responses and locations of the revisions: We thank the reviewer for pointing this out. We have changed P to be italicized.

Comment 4:  The units of Bacillus subtilis in the text are written in many different forms, such as CFU g-1, cfu/kg, CFU/g. Please standardize your writing.

Authors'responses and locations of the revisions: We thank the reviewer for pointing this out. We have modified and standardized to CFU/ g in paper.

Comment 5:  The manuscript cites a large number of references from 10 years ago, and the author should cite the literature from the past 3-5 years.

Authors'responses and locations of the revisions: We appreciate the reviewer’s insightful comment and have made change to new the references, as possible.

  1. The language of the manuscript needs professional revision.

Authors'responses and locations of the revisions: We appreciate the reviewer’s insightful comment and will take this into consideration.

Reviewer 2 Report

Dear Authors, the presented manuscript concerns a current and necessary topic. However, I believe that you were guided by a research hypothesis that was missing from the manuscript. I made a few other comments in the review which I believe will enrich the manuscript and make it more complete.

Line 48: I think it is better to formulate the sentence that poultry meat can prevent excessive build-up of cholesterol in the blood of poultry meat consumers.

Line 78: Please propose a research hypothesis.

Line 103: Please indicate the composition of the feed mixture. It is an important element of research when assessing production indicators.

Line 125-142: Please specify how the birds are churned. Was there the approval of the appropriate commission for these research?

Line 149-152: What analytical techniques were used to assess blood indicators? Please mention this in the methodology.

Line 179: Please, use a space.

Line 181-182: The table title should be written above the table, not below it. Please edit the appropriate font for the text in the table. Which SI unit is valid in the table? Are they grams, decagrams, or maybe kilograms? Please complete this.

Table 3 and Table 4: Please use appropriate units for all indicators, consistent with the SI system.

Discussion: I suggest trying to better explain the mechanisms of the changes that have occurred, which are manifested by a change in the metabolism of chickens. I suggest using a bit newer, additional literature dealing with this topic. You can use:

1.         Mohamed TM, Sun W, Bumbie GZ, et al. Feeding Bacillus subtilis ATCC19659 to Broiler Chickens Enhances Growth Performance and Immune Function by Modulating Intestinal Morphology and Cecum Microbiota. Front Microbiol. 2022;12:798350. Published 2022 Feb 22. doi:10.3389/fmicb.2021.798350

2. Krauze, M.; Abramowicz, K.; Ognik, K. The effect of addition of probiotic bacteria (Bacillus subtilis or Enterococcus faecium) or phytobiotic containing cinnamon oil to drinking water on the health and performance of broiler. Ann. Anim. Sci. 2020, 20, 191–205

Author Response

Section: "Responses to the Comments by reviewer 2"

Dear Authors, the presented manuscript concerns a current and necessary topic. However, I believe that you were guided by a research hypothesis that was missing from the manuscript. I made a few other comments in the review which I believe will enrich the manuscript and make it more complete.

We thank you very much for the comments and suggestions, giving us a chance to revise and improve the manuscript.

Comment 1: Line 48: I think it is better to formulate the sentence that poultry meat can prevent excessive build-up of cholesterol in the blood of poultry meat consumers.

Authors'responses and locations of the revisions: We thank the reviewer for pointing this out. We have made changes in line 47 to 50 on page 2.

Comment 2: Line 78: Please propose a research hypothesis.

Authors'responses and locations of the revisions: We thank the reviewer for pointing this out. We have modified to clear our hypothesis in line 86 to 93 on page 2.

Comment 3: Line 103: Please indicate the composition of the feed mixture. It is an important element of research when assessing production indicators.

Authors'responses and locations of the revisions: Thanks. Owing to your suggestion, we added the feed composition in supplementary data because we published it in our previous paper and we found it difficult to present it in our manuscript because of replication.

Comment 4: Line 125-142: Please specify how the birds are churned. Was there the approval of the appropriate commission for these researches?

Authors'responses and locations of the revisions: Thanks. Owing to your suggestion, we took approval from the appropriate commission. We sent approval to the journal editor's office.

Comment 5: Line 149-152: What analytical techniques were used to assess blood indicators? Please mention this in the methodology.

Authors'responses and locations of the revisions: Thanks. Owing to your suggestion, we sent serum blood to the commercial scientific lab to measure biochemical blood, and the lab used kits from the Nanjing Jiancheng Bioengineering Institute, Nanjing, China. We have sent the codes for each type of kit that the lab used in supplementary data and some photos of the lab as a supplementary file.

Comment 6: Line 179: Please, use a space.

Authors'responses and locations of the revisions: Thanks. Owing to your suggestion, we have removed the space

Comment 7: Line 181-182: The table title should be written above the table, not below it. Please edit the appropriate font for the text in the table. Which SI unit is valid in the table? Are they grams, decagrams, or maybe kilograms? Please complete this.

Authors'responses and locations of the revisions: Thanks. Owing to your suggestion, we modified the table title to the above and edit the appropriate font for the text in the table. We modified the SI unit (grams).

Comment 8: Table 3 and Table 4: Please use appropriate units for all indicators, consistent with the SI system.

Authors'responses and locations of the revisions: We thank the reviewer for pointing this out. We checked the units and we found all indicators use appropriate units

Comment 9: Discussion: I suggest trying to better explain the mechanisms of the changes that have occurred, which are manifested by a change in the metabolism of chickens. I suggest using a bit newer, additional literature dealing with this topic. You can use:

Authors'responses and locations of the revisions: Thanks. Owing to your suggestion.  We explained more mechanisms which change with the administration of  Bacillus subtilis and we were grateful for presenting these two papers.

Reviewer 3 Report

Dear authors,

I’ve had the opportunity to assist you with your manuscript.  I think  it is a good work and of great global interest. After reviewing your work I have made some considerations to further enhance your paper.

Please review and revise if you agree.

Regarding the main sections, kindly note as follows:

title:

delete the number of the strain, insert it only in materials and methods section to describe the strain. In the rest of the manuscript comes across as unnecessary and confusing, also because in your study you used a single strain at different concentrations.

Change also with a synonymous “carcass traits” it is unclear.

Introduction:

 lane 53 to 60 are written in bold type, fix it.

Lane 67: the verb plant is not appropriate for the spores, change it.

Lane 67 to 71 the period is unclear, revise it.

Lane 79 better concentrations and not “doses” when referd to a bacterial strain

Materials and methods

Lane 96 the period is unclear. Try to explain better.

About the section of blood analysis the type of analysis performed is not well described, please improve it and add the description of the kit you used for.

Results

Some results are very interesting.

Please correct the word Fabricious with the capital letter.

About paragraph 3.4  what the authors mean by serum kidney?  Please change the title of this paragraph.

Discussion

The authors show in their study an increased size of the bursa of Fabricious in group B.Sut. 3 at day 21, compared to the control, however the later control at 42 days there is a downsizing.

 what are your hypotheses in this regard?

Please improve your discussion about this point.

Best regards

Author Response

Section: "Responses to the Comments by reviewer 3"

Dear authors,

I’ve had the opportunity to assist you with your manuscript.  I think it is good work and of great global interest. After reviewing your work I have made some considerations to further enhance your paper.

Please review and revise if you agree.

Regarding the main sections, kindly note as follows:

 We appreciate your approval of our manuscript as well as your thoughtful assessment, and we hope that we have addressed all of your suggestions.

Title:

Comment 1: delete the number of the strain, insert it only in materials and methods section to describe the strain. In the rest of the manuscript comes across as unnecessary and confusing, also because in your study you used a single strain at different concentrations.

Authors'responses and locations of the revisions: Thanks. Owing to your suggestion, we modified title and changed in all paper according to your suggestion.

Comment 2: Change also with a synonymous “carcass traits” it is unclear.

Authors'responses and locations of the revisions: Thanks. Owing to your suggestion, we changed to organ weight. 

Introduction:

Comment 3:  lane 53 to 60 are written in bold type, fix it.

Authors'responses and locations of the revisions: We thank the reviewer for pointing this out. The Lines are changed and fixed.

Comment 4:  Lane 67: the verb plant is not appropriate for the spores, change it.

Authors'responses and locations of the revisions: We thank the reviewer for pointing this out. We have changed in line 76 on page 2.

Comment 5:  Lane 67 to 71 the period is unclear, revise it.

Authors'responses and locations of the revisions: We thank the reviewer for pointing this out. We have removed the word (Please see line 78 Page 2 the revised version). 

Comment 6:  Lane 79 better concentrations and not “doses” when referd to a bacterial strain.

Authors'responses and locations of the revisions: We thank the reviewer for pointing this out.  Really we have modified and changed this sentence for rewriting the hypotheses.  

Materials and methods

Comment 7: Lane 96 the period is unclear. Try to explain better.

Authors'responses and locations of the revisions: We thank the reviewer for pointing this out. We have modified and rewrote it to be clearer (Please see line 112 to 116 Page 3 of the revised version).

Comment 8: About the section of blood analysis the type of analysis performed is not well described, please improve it and add the description of the kit you used for.

Authors'responses and locations of the revisions: We thank the reviewer for pointing this out. we sent serum blood to the commercial scientific lab to measure biochemical blood, and the lab used kits from the Nanjing Jiancheng Bioengineering Institute, Nanjing, China. We have sent the codes for each type of kit that the lab used in supplementary data and some photos of the lab as supplementary file.

Results

Some results are very interesting.

Comment 9: Please correct the word Fabricious with the capital letter.

Authors'responses and locations of the revisions: We thank the reviewer for pointing this out. We have changed to Fabricious in paper.

Comment 10: About paragraph 3.4 what the authors mean by serum kidney?  Please change the title of this paragraph.

Authors'responses and locations of the revisions: We thank the reviewer for pointing this out. We have changed to kidney function.

Discussion

Comment 11: The authors show in their study an increased size of the bursa of Fabricious in group B.Sut. 3 at day 21, compared to the control, however the later control at 42 days there is a downsizing.          

Authors'responses and locations of the revisions: We thank the reviewer for pointing this out.  The size of the bursa of Fabricius was decreased because broiler body weight in increased at 42 days (Agboola et al., 2015).

Comment 12:  what are your hypotheses in this regard?

Please improve your discussion about this point. 

Authors'responses and locations of the revisions: We thank the reviewer for pointing this out. We modified the hypotheses in our paper.

Agboola, A. F., B. R. O. Omidiwura, O. Odu, I. O. Popoola, and E. A. Iyayi. "Effects of organic acid and probiotic on performance and gut morphology in broiler chickens." South African Journal of Animal Science 45, no. 5 (2015): 494-501.
